# SARS-CoV-2 seroepidemiology in Cape Town, South Africa, and implications for future outbreaks in low-income communities

**Hannah Hussey**[1,2,3]*, **Helena Vreede**[4,5], **Mary-Ann Davies**[1,3,6], **Alexa Heekes**[1,6], **Emma Kalk**[3,6], **Diana Hardie**[5,7], **Gert van Zyl**[5,8], **Michelle Naidoo**[5,7,8], **Erna Morden**[1,3], **Jamy-Lee Bam**[1], **Nesbert Zinyakatira**[1,3], **Chad M. Centner**[5], **Jean Maritz**[8,9], **Jessica Opie**[5,10], **Zivanai Chapanduka**[5,11], **Hassan Mahomed**[2,12], **Mariette Smith**[1,6], **Annibale Cois**[3,12], **David Pienaar**[13], **Andrew D. Redd**[14,15], **Wolfgang Preiser**[5,8], **Robert Wilkinson**[16,17,18], **Andrew Boulle**[1,3,6], **Nei-yuan Hsiao**[5,7]

1 Health Intelligence Directorate, Western Cape Government: Health, Cape Town, South Africa, 2 Metro Health Services, Western Cape Government: Health, Cape Town, South Africa, 3 School of Public Health, University of Cape Town, Cape Town, South Africa, 4 Division of Chemical Pathology, University of Cape Town, Cape Town, South Africa, 5 National Health Laboratory Service, Cape Town, South Africa, 6 Centre for Infectious Disease Epidemiology and Research, School of Public Health, University of Cape Town, Cape Town, South Africa, 7 Division of Medical Virology, University of Cape Town, Cape Town, South Africa, 8 Division of Medical Virology, Stellenbosch University, Cape Town, South Africa, 9 PathCare Reference Laboratory, Cape Town, South Africa, 10 Division of Haematology, University of Cape Town, Cape Town, South Africa, 11 Division of Haematology, Stellenbosch University, Cape Town, South Africa, 12 Division of Health Systems and Public Health, Department of Global Health, Faculty of Medicine and Health Sciences, Stellenbosch University, Cape Town, South Africa, 13 Rural Health Services, Western Cape Government: Health, Cape Town, South Africa, 14 Department of Medicine, Johns Hopkins University School of Medicine, Baltimore, Maryland, United States of America, 15 Division of Intramural Research, National Institute of Allergy and Infectious Diseases, National Institutes of Health, Bethesda, Maryland, United States of America, 16 Wellcome Centre for Infectious Disease Research in Africa, Institute of Infectious Disease and Molecular Medicine, University of Cape Town, Cape Town, South Africa, 17 The Francis Crick Institute, London, United Kingdom, 18 Department of Infectious Diseases, Imperial College London, London, United Kingdom

* hannah.hussey@westerncape.gov.za

## Abstract

In low- and middle-income countries where SARS-CoV-2 testing is limited, seroprevalence studies can help describe and characterise the extent of the pandemic, as well as elucidate protection conferred by prior exposure. We conducted repeated cross-sectional serosurveys (July 2020–November 2021) using residual samples from patients from Cape Town, South Africa, sent for routine laboratory studies for non-COVID-19 conditions. SARS-CoV-2 anti-nucleocapsid antibodies and linked clinical information were used to investigate: (1) seroprevalence over time and risk factors associated with seropositivity, (2) ecological comparison of seroprevalence between subdistricts, (3) case ascertainment rates, and (4) the relative protection against COVID-19 associated with seropositivity and vaccination statuses. Among the subset sampled, seroprevalence of SARS-CoV-2 in Cape Town increased from 39.19% (95% confidence interval [CI] 37.23–41.19) in July 2020 to 67.8% (95%CI 66.31–69.25) in November 2021. Poorer communities had both higher seroprevalence and COVID-19 mortality. Only 10% of seropositive individuals had a recorded positive SARS-CoV-2 test. Using COVID-19 hospital admission and death data at the Provincial Health Data Centre, antibody positivity before the start of the Omicron BA.1 wave (28

**Data Availability Statement:** The underlying data are routinely collected patient records that have

been de-identified and pseudo-anonymised. The patients have not consented to these data being part of publicly accessible repositories considering the inherent risks of re-identification. The Western Cape Department of Health and Wellness evaluates research proposals for all research in the public health sector in the province, subject to standard research ethics, government approval and data governance prescripts. This includes those that draw on routine datasets like the current study. For more information, including to start the data access request process, please email health. research@westerncape.gov.za.

**Funding:** The study is funded by National Health Laboratory service, Western Cape Department of Health, Wellcome Trust, and in part by the Division of Intramural Research, NIAID, NIH. RJW is supported by the Francis Crick Institute which receives funding from Cancer Research UK (FC0010218), Medical Research Council (FC0010218), and Wellcome (FC0010218). He also received funding from Wellcome (203135,222754). The funders had no role in the study design, collection and analysis of data and decision in submission for publication.

**Competing interests:** The authors have declared that no competing interests exist.

November 2021) was strongly protective for severe disease (adjusted odds ratio [aOR] 0.15; 95%CI 0.05–0.46), with additional benefit in those who were also vaccinated (aOR 0.07, 95%CI 0.01–0.35). The high population seroprevalence in Cape Town was attained at the cost of substantial COVID-19 mortality. At the individual level, seropositivity was highly protective against subsequent infections and severe COVID-19 disease. In low-income communities, where diagnostic testing capacity is often limited, surveillance systems dependent on them will underestimate the true extent of an outbreak. Rapidly conducted seroprevalence studies can play an important role in addressing this.

## Introduction

Most low- and middle-income countries, experienced multiple waves of COVID-19 cases in the first two years of the pandemic, but due to a relative lack of access to testing, the reported case numbers are a gross underestimate of the true scale of infection [1]. In South Africa, excess deaths from natural causes in 2020–2021 was 295 135, compared to the official COVID-19 reported deaths of 93 816 [2]. While not all these deaths were directly due to SARS-CoV-2 infection, the fact that excess deaths were three times higher than reported deaths highlights the high levels of case under-ascertainment. This situation is made more complex by differential levels of case under-ascertainment in different communities, with low income communities having less access to testing [3]. Barriers to testing are both at the individual level, including health literacy, trade-offs of knowing results and personal costs of time and out-of-pocket expenses to get tested, as well as at the health system level, with limited health system preparedness and laboratory capacity in poorer settings [4].

Recent serosurvey studies in South Africa [5–10] and Africa [11,12] have demonstrated high levels of sero-positivity from infection or vaccination two years after the emergence of SARS-CoV-2. However, due to their cross-sectional design, few large seroprevalence studies have been able to estimate the degree of protection conferred by seropositivity beyond ecological comparisons [5].

A serosurvey that is nested within a population cohort can provide insight into the relative protection provided by prior infection and/or vaccination. This is particularly useful in South Africa's complex immune landscape where many individuals have hybrid immunity from both prior infection with different variants as well as vaccination [13]. In this study we use serology with linked infection and outcome data from a population-level health database to describe in detail the longitudinal epidemiology of SARS-CoV-2 in Cape Town, South Africa, across three SARS-CoV-2 waves between March 2020 and March 2022. These waves peaked in June 2020, and January, August and December 2021, and were caused by the ancestral SARS-CoV-2, the Beta, Delta and Omicron (BA.1/BA.2 sub-lineages) variants, respectively (Fig 1) [14].

We aimed to assess, firstly SARS-CoV-2 seroprevalence trends, including risk factors associated with seropositivity, an ecological comparison of seroprevalence rates (using subdistricts of residence) and their differing case ascertainment rates, and then, lastly, the relative protection conferred by different combinations of prior infection and vaccination against future COVID-19 infection and severe disease.

While SARS-CoV-2 antibody positivity does wane with time [15]), by assessing the seropositivity between waves, and comparing that to the recorded infections in the preceding and subsequent waves, we are able to assess the impact of seropositivity beyond what a more traditional cross-sectional study would be able to do.

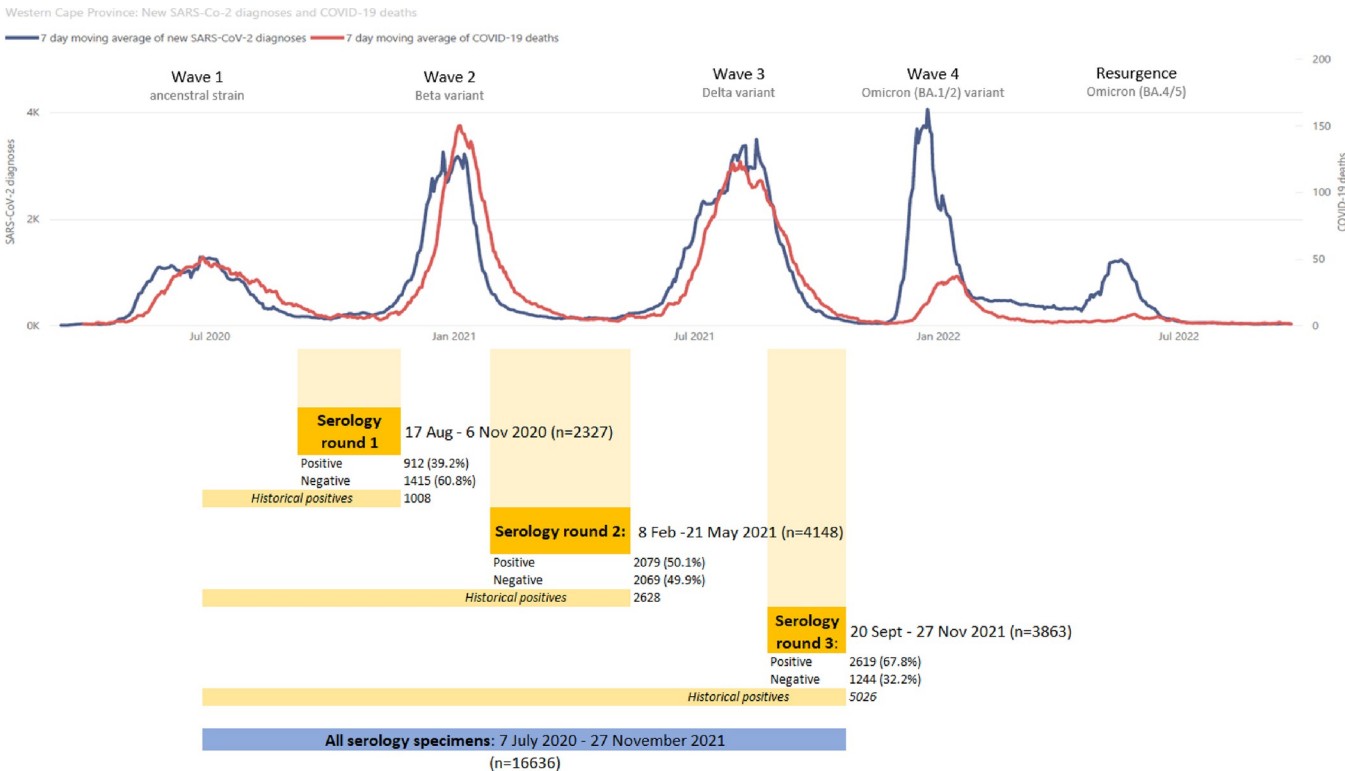

**Fig 1. Line graph of new SARS-CoV-2 diagnoses and COVID-19 deaths in both the public and private sectors of the Western Cape Province (March 2020 to August 2022), combined with the time periods of the included serology specimens to describe the relationship of the four COVID-19 waves relative to the serology testing rounds.**

While the World Health Organisation has recently called an end to the global health emergency of COVID-19 [16], and interest in and concern about the virus has understandably waned, understanding how a novel pathogen spreads through low- and middle-income countries like South Africa, which communities are most at risk, and what the limitations and biases of using routine diagnostic systems for surveillance in these setting are, remain critical.

## Methods

### Source population

The city of Cape Town in South Africa had an estimated 4.6 million population in 2020 residing in eight subdistricts. Approximately 75% of the population does not have medical insurance and is dependent on public sector health services [13]. The routine laboratory diagnostic testing for the public sector is provided by the National Health Laboratory Service (NHLS) laboratories.

We conducted three rounds of cross-sectional serosurveys in Cape Town. We selected consecutive convenience routine plasma samples collected for the purposes of 1) antenatal blood grouping, 2) HIV viral load monitoring, 3) glycated haemoglobin testing and 4) various markers for paediatric in-patients and outpatients across various tiers of health facilities. Samples were retrieved from NHLS between April 2020 and November 2021. In that time period, over 600 000 specimens were processed by the NHLS in the Cape Town Metro, including 193 993 HbA1c specimens and 412 539 HIV VL specimens. After the first round, antenatal sampling were replaced by glycated haemoglobin samples to minimise sex and age biases. Consecutive

samples per sub-district per sampling interval were selected, until a predetermined number, subject to available budget, was met. Samples were excluded if on visual inspection they were found to be grossly haemolysed or likely to have insufficient volume.

## Assays

Prior infection with SARS-CoV-2 was assessed through detection of total anti-nucleocapsid (anti-N) antibody using the Elecsys anti-N SARS-CoV-2 assay (Roche, Geneva, Switzerland) on the Roche Cobas e601 platform according to manufacturer's instructions. The assay is widely used for population seroprevalence studies. Its performance has been evaluated in South Africa where the specificity was 100% and sensitivity was 89% for detection of PCR-confirmed infection >30 days post diagnosis [17].

## Linkage of serology results to clinical data

The Western Cape Provincial Health Data Centre (PHDC) links patient administrative records, laboratory, pharmacy and vaccination data from routine electronic clinical information systems used in all public sector health facilities through a unique patient identifier [18]. Since March 2020, the PHDC has collated information on all SARS-CoV-2 testing and related hospital admissions as part of routine government surveillance activities. With limited testing capacity in the Western Cape, SARS-CoV-2 testing was limited to those with severe disease, old age or medical comorbidities, particularly during the peaks of the waves, although a slight increase in testing capacity was noted from October 2020 when rapid antigen tests became available [19].

Serology results were linked at the PHDC to create an integrated dataset describing a cohort of patients with all COVID-19 testing, vaccination status, admission and mortality data, patient demographics, subdistrict of last primary care visit, and known co-morbidities (HIV, tuberculosis history, diabetes, chronic kidney disease, hypertension, asthma/COPD). Only data scientists employed at the PHDC (AH, JB, MS) have access to identifiers that could identify participants. Their access is authorised by the Western Cape Provincial Department of Health and Wellness in their capacity as data scientists employed at the PHDC, for the purposes of data integration and participant data linkage. All access to these identifiers is logged and protocols are in place to ensure only de-identified data can be released for analytical purposes. The linked dataset for this study was fully de-identified prior to being downloaded and accessed by HH for analysis on 24 May 2022.

Vaccination status was determined through the South African Electronic Vaccination Data System (EVDS). Vaccination status, which was defined as either single or double dose, was assessed at time of COVID-19 diagnosis, or at the peak of the wave if the individual did not test positive for COVID-19 during the wave of interest (3 January 2021 for second wave, 1 August 2021 for third wave and 19 December 2021 for fourth wave). Single dose vaccination could be with either Pfizer–BioNTech (BNT162b2) or Janssen/Johnson & Johnson (Ad26. COV2.S; J&J), while two dose vaccination was defined as being more than 14 days after receipt of any two doses of COVID-19 vaccines. Both of these vaccinations are based on viral spike sequences and therefore do not provoke an anti-N antibody response.

From 17 May 2021, primary vaccination with either BNT162b2 (2 doses) or Ad26.COV2.S (single dose) was available in South Africa, first to adults aged 60 years and above, and then progressively to younger age groups, with all individuals aged 12 years and older being eligible from 20 October 2021 [20]. Booster vaccinations were not assessed as these only became widely available in South Africa from March 2022, when the fourth wave was already ending [21], and so insufficient numbers of boosted individuals were available for this analysis.

### Descriptive and ecological analyses

Sero-positivity and 95% confidence estimates were calculated for the serology study population, stratified by age, sex, comorbidity and presumed subdistrict of residence based on location of last primary health care facility visit. Logistic regression to determine the odds ratio for having positive serology was also calculated, adjusted for age, sex, serology round and subdistrict of residence. To assess the impact of socio-economic status on SARS-CoV-2 infection and outcome, subdistrict seroprevalence estimates were correlated with subdistrict household income data and COVID-19 death rates. The proportion of low-income households in each of the subdistricts was determined using the latest available Census data from 2011 [22]. Age-standardised COVID-19 death rates by subdistrict were determined using previously described methodology [23].

### Individual linked analyses

For the linked analysis to estimate protection associated with prior infection and/or vaccination, three separate analyses for each wave were conducted. All serology specimens preceding the wave of interest, as well as any additional historical positive specimens, were included in the analysis of that wave (Fig 1)—e.g., the second round of serology testing between 8 February and 21 May 2021, before the third wave, was used to look at the outcomes in the third wave. Prior infection was defined as individuals with positive anti-N serology in any period prior to the wave of interest, i.e., "historical" positive specimens were included in analyses of later waves, while only negative specimens from the serology round immediately preceding the wave could be included (as previously negative patients could have become infected subsequently, but before the wave of interest).

We categorised linked serology specimens according to their anti-N result and vaccination status. We used logistic regression to assess the outcomes of confirmed symptomatic infection (i.e., having a documented positive SARS-CoV-2 test in the Western Cape), being a non-severe case (i.e., a positive SARS-CoV-2 test but not admitted to hospital nor deceased), or a severe case (a positive SARS-CoV-2 test and admitted or deceased), adjusted for age, sex, subdistrict of residence and known comorbidities. We considered admissions within ±14 days of a positive antigen or PCR test as COVID-19 related and deaths up to 28 days after a positive COVID-19 test or within 14 days of discharge following a COVID-19 related admission to be COVID-19-related deaths.

### Ethics statement

The study was approved by the Human Research Ethics Committees of the University of Cape Town (HREC REF 449/2020) and Stellenbosch University (N20/08/051). Institutional approval was obtained from the National Health Laboratory Service and the Western Cape Department of Health. Individual informed consent was waived as residual specimens and clinical data collected during the course of routine care were used in this retrospective study, and the analysis was performed on a deidentified dataset.

## Results

### Descriptive and ecological analyses results

Our study tested a total of 16 636 residual plasma samples across the first four waves of the SARS-CoV-2 pandemic in Cape Town.

Of the 16 636 specimens included, 68.5% were from females. 8.4% and 13.2% were from those aged 14 years and under and 65 years and older, respectively. This is in comparison to

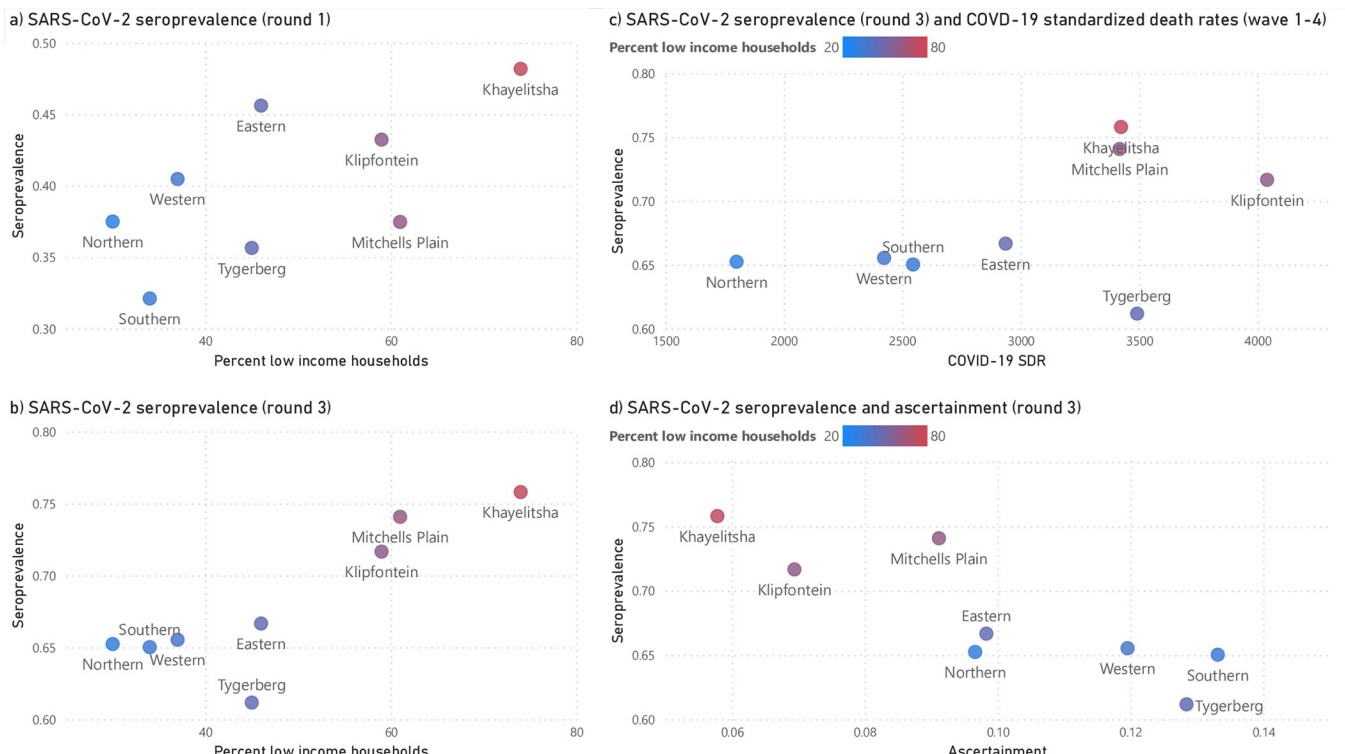

**Fig 2. Anti-N seroprevalence and 95% confidence interval, grouped by serology round and age category, sex, comorbidity or subdistrict of residence.**

the 2022 Cape Town Census results (not fully released yet) which found that the population is 51.7% female; 22.4% were aged 14 and under and 6.7% aged 65 years and older [24]. 37.1% of our samples were taken from diabetic patients and 41.6% were taken from people living with HIV. To note, the HIV prevalence in the Western Cape was estimated to be 10.3% in adults aged 15–49 years in 2018 [25], and the prevalence of diabetes in South African adults could be as high as 22% if undiagnosed diabetes is taken into account [25,26].

Of 16 636 specimens, 10 338 could be grouped into three rounds of serology testing in the inter-wave periods (Fig 1). Anti-N seroprevalence increased from 39.19% (95% confidence interval [95%CI] 37.23–41.19%) in the first round (i.e., after the first wave, 17 August to 6 November 2020), to 50.12% (95%CI 48.60–51.64%) in the second round (8 February to 21 May 2021), and to 67.80% (95%CI 66.31–69.25%) in the third round before the start of the fourth wave (20 September to 27 November 2021).

There was heterogeneity in seroprevalence across age, sex, comorbidities and subdistrict of residence (Fig 2). Higher seroprevalence was observed in individuals aged 17–49, compared to other age groups. Women had higher seroprevalence than men. People living with HIV also had higher seroprevalence than those with diabetes. These risk factors for seropositivity persisted after multivariable adjustment (Table 1).

While seroprevalence increased with time across Cape Town, there was a strong correlation between anti-N positivity and subdistrict poverty levels, with the poorest subdistrict, Khayelitsha, consistently having the highest seroprevalence (Fig 3A and 3B). This resulted in differing wave patterns in the different subdistricts (Fig 4). The Northern subdistrict, the most affluent in the Cape Town Metro, had a relatively small first wave, but increasingly large second and third waves. Khayelitsha, the poorest subdistrict, in contrast, had a very large first wave and then second and third waves of decreasing size. Fig 3C describes how low-income subdistricts,

**Table 1. Logistic regression for the outcome of having anti-N positive serology.**

| Logistic regression for outcome of having positive anti-N serology | | | | |
| --- | --- | --- | --- | --- |
| (n = 10338) | | | | |
| | | aOR | (95% CI) | |
| Sex | Female | Ref | | |
| | Male | 0.85 | 0.78 | 0.93 |
| | | | | |
| Age category | 0–16 years | Ref | | |
| | 17–49 years | 1.24 | 1.03 | 1.50 |
| | 50–69 years | 0.91 | 0.75 | 1.12 |
| | ≥ 70 years | 0.53 | 0.42 | 0.67 |
| | | | | |
| Comorbidity (the reference group here is the absence of that specific comorbidity) | Diabetes | 1.17 | 1.05 | 1.31 |
| | HIV positive | 1.45 | 1.29 | 1.62 |
| | Tuberculosis (ever) | 0.85 | 0.76 | 0.96 |
| | Hypertension | 1.05 | 0.95 | 1.17 |
| | Chronic obstructive pulmonary disease | 0.79 | 0.69 | 0.89 |
| | Chronic kidney disease | 0.91 | 0.78 | 1.07 |
| | | | | |
| Serology round | 1 | Ref | | |
| | 2 | 1.97 | 1.77 | 2.20 |
| | 3 | 3.92 | 3.49 | 4.40 |
| | | | | |
| Subdistrict of residence | Eastern | Ref | | |
| | Khayelitsha | 1.53 | 1.28 | 1.83 |
| | Klipfontein | 1.32 | 1.10 | 1.58 |
| | Mitchells Plain | 1.10 | 0.93 | 1.31 |
| | Northern | 0.92 | 0.75 | 1.14 |
| | Southern | 0.93 | 0.78 | 1.11 |
| | Tygerberg | 0.82 | 0.69 | 0.99 |
| | Western | 1.01 | 0.85 | 1.20 |

which had the highest seroprevalence rates, also had the highest standardised COVID-19 death rates.

The case ascertainment rate, that is the proportion of individuals with antibody positive serology results who had a positive SARS-CoV-2 test recorded at any time before their positive serology, remained relatively stable over time (0.12, 0.12 and 0.10 in serology rounds 1,2 and 3 respectively–see Table 2). Case ascertainment displayed marked variation by subdistrict, with the poorer subdistricts having lower ascertainment rates (Fig 3D). For example, in Khayelitsha, only 6% of cases were ascertained in both the first and third round of serology testing, compared to the Western subdistrict where 23% and 12% of cases were ascertained, respectively.

## Serology and linked subsequent infection/outcome results

For the linked analysis, 8 184 linked serology episodes available to assess outcomes in the fourth wave and 6 306 and 3 215 specimens for the third and second waves, respectively. The distribution of antibody results and vaccination status, as well as outcomes, is shown in Table 3. As the vaccination programme in the general population only began in May 2021, everyone was unvaccinated in the second wave. Prior to the start of the fourth wave, 29.06% of

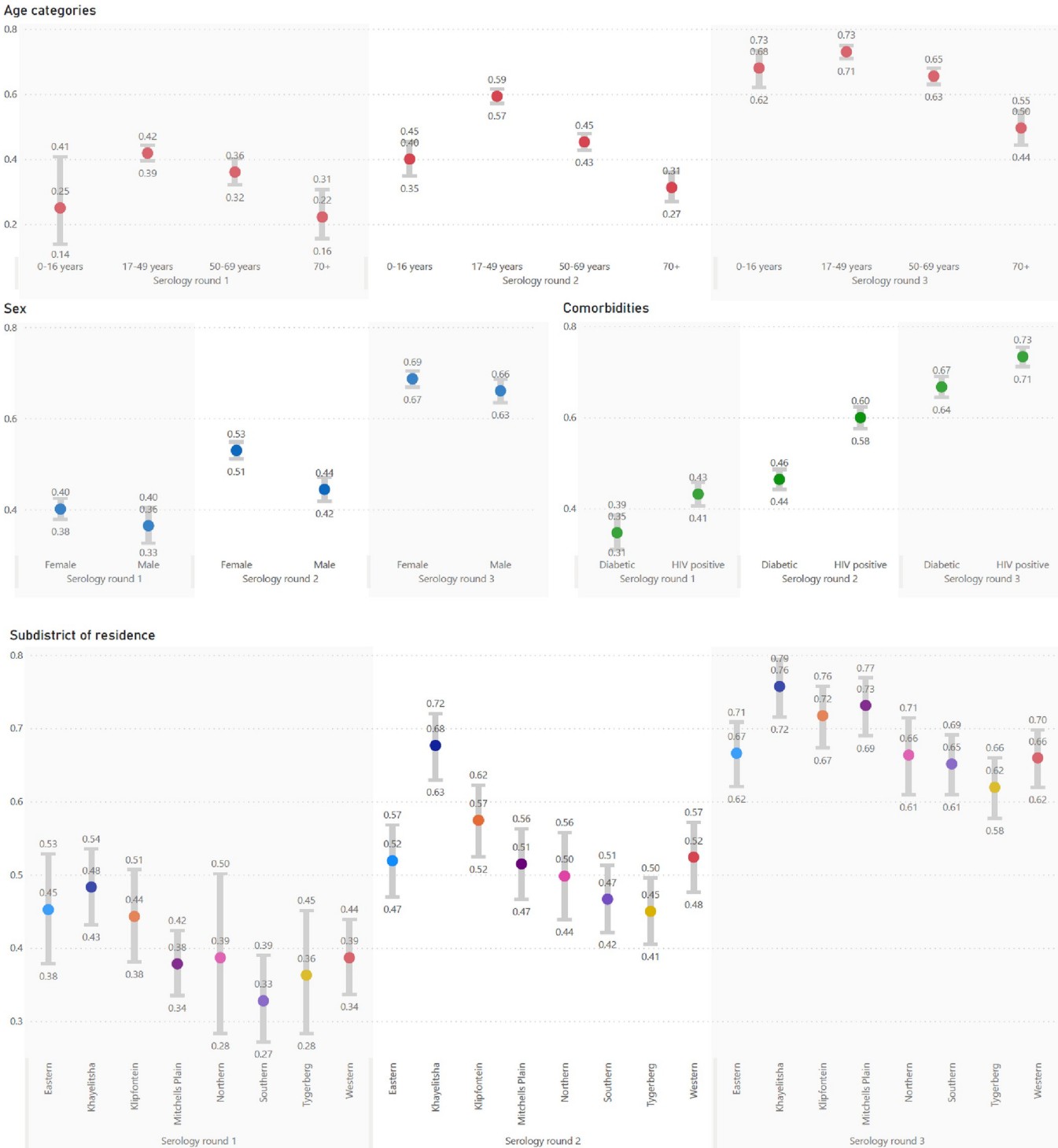

**Fig 3.** Anti-N seroprevalence scatter plots describing the association between (a) seroprevalence in serology round 1 (August–November 2020) with proportion of low-income households, (b) as above but for serology round 3 (September–November 2021), (c) seroprevalence in round 3 and age and sex-standardized COVID-19 deaths rates (SDR), combined for waves 1–4, and (d) seroprevalence in round 3 compared to the case ascertainment rate. In all the scatter plots, subdistrict is the unit of measurement, and is colour coded according to proportion of low-income households.

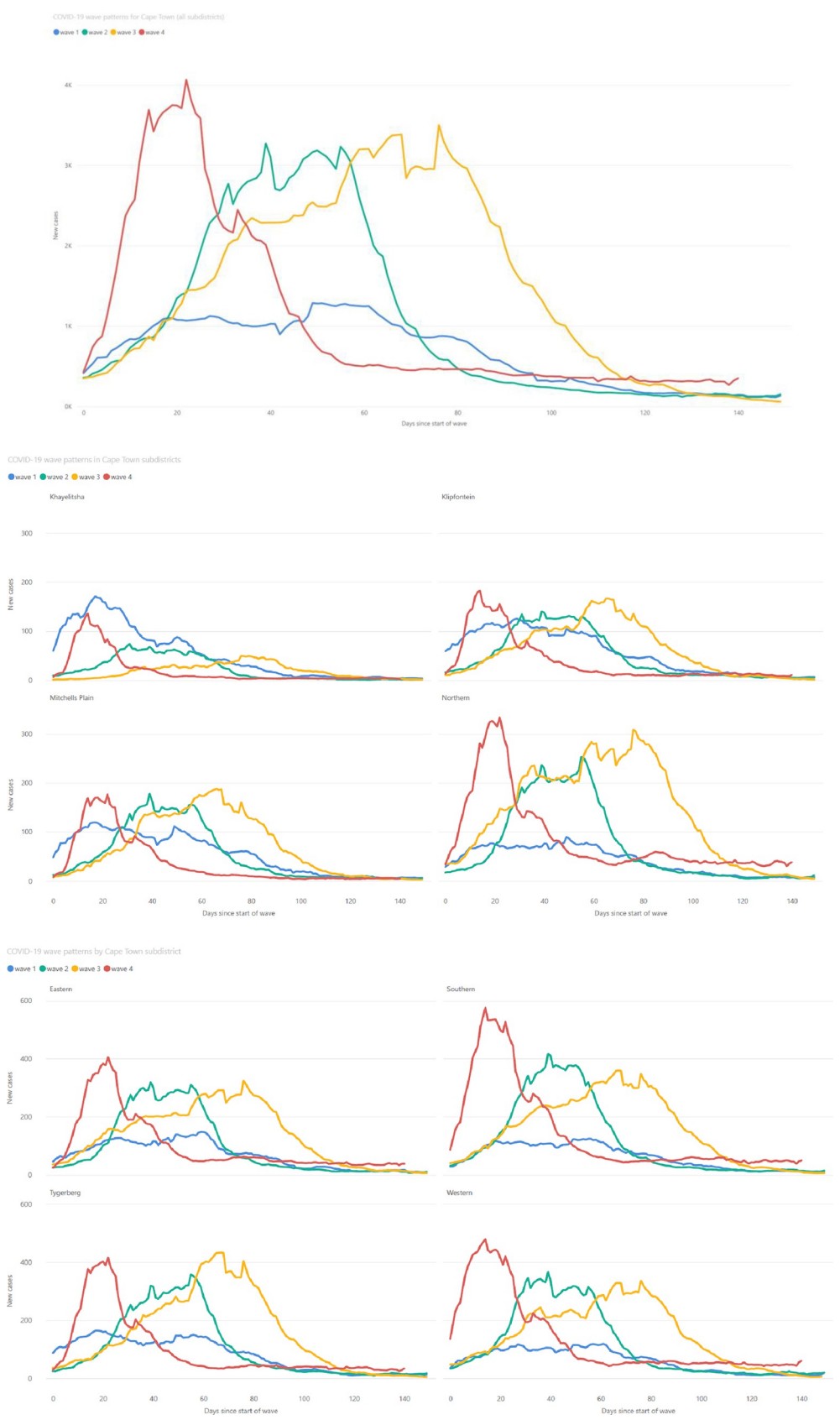

**Fig 4. 7 day moving average of new SARS-CoV-2 cases, in both the public and private sector, in the City of Cape Town as a whole and in the eight subdistricts of Cape Town, showing the differing patterns of COVID-19 waves.**

individuals with serology specimens taken 20 September to 27 November were fully vaccinated against SARS-CoV-2, with an additional 11.95% being partially vaccinated. In all waves, Anti-N antibody presence was associated with protection against subsequent confirmed symptomatic infection or having severe disease. Protection against severe disease was highest in the Delta wave (aOR of 0.02; 95%CI 0.00–0.11 for antibody positive vs negative in unvaccinated individuals), and lowest in the Omicron wave (aOR of 0.15; 95%CI 0.05–0.35). Vaccination (2 doses) in anti-N positive individuals further strengthened this protection against severe disease in the Omicron wave (aOR 0.07, 95%CI 0.01–0.35 for vaccinated anti-N positive vs unvaccinated anti-N negative).

## Discussion

Our seroprevalence data following three waves of the SARS-CoV-2 epidemic in Cape Town indicate that despite public health restrictions, the burden of infection in the community was high and that anti-N seropositivity as a marker of past infection was strongly protective against severe disease during a subsequent infection, irrespective of variant.

**Table 2. Case ascertainment rate for each round of serology testing, as well by subdistrict of residence, determined by calculating the proportion of positive anti-N antibody results, that had a laboratory confirmed SARS-CoV-2 diagnosis at any time prior to their serology result.**

| Proportion of positive anti-N specimens that had a diagnosed SARS-CoV-2 specimen at any time prior to serology testing | | | | |
|---|---|---|---|---|
| | | **Proportion ascertainment** | **95%CI** | |
| **Combined for serology round 1 = 0.12 (95%CI 0.10–0.14)** | **Eastern** | **0** | **(no observations)** | |
| | Khayelitsha | 0.06 | 0.04 | 0.11 |
| | Klipfontein | 0.12 | 0.07 | 0.20 |
| | Mitchells Plain | 0.12 | 0.08 | 0.18 |
| | Northern | 0.07 | 0.02 | 0.24 |
| | Southern | 0.22 | 0.14 | 0.32 |
| | Tygerberg | 0.09 | 0.03 | 0.22 |
| | Western | 0.23 | 0.16 | 0.30 |
| Combined for serology round 2 = 0.12 (95%CI 0.11–0.14) | Eastern | 0.08 | 0.05 | 0.12 |
| | Khayelitsha | 0.08 | 0.05 | 0.11 |
| | Klipfontein | 0.10 | 0.07 | 0.14 |
| | Mitchells Plain | 0.07 | 0.04 | 0.12 |
| | Northern | 0.08 | 0.05 | 0.14 |
| | Southern | 0.14 | 0.10 | 0.20 |
| | Tygerberg | 0.12 | 0.08 | 0.17 |
| | Western | 0.12 | 0.08 | 0.17 |
| Combined for serology round 3 = 0.10 (95%CI 0.09–0.11) | Eastern | 0.10 | 0.07 | 0.14 |
| | Khayelitsha | 0.06 | 0.04 | 0.09 |
| | Klipfontein | 0.07 | 0.05 | 0.10 |
| | Mitchells Plain | 0.09 | 0.07 | 0.13 |
| | Northern | 0.10 | 0.06 | 0.15 |
| | Southern | 0.13 | 0.10 | 0.17 |
| | Tygerberg | 0.13 | 0.10 | 0.17 |
| | Western | 0.12 | 0.09 | 0.16 |

**Table 3. Linked serology analysis, showing the numbers of serology specimens and outcomes in each risk category, as well as logistic regression for the outcome of being a SARS-CoV-2 case, with and without severe disease.** This analysis was adjusted for age category, sex, known comorbidities and subdistrict of residence.

| | | n | Cases | Admission or death | All cases | | | Non-severe cases | | | Admissions or death | | |
|---|---|---|---|---|---|---|---|---|---|---|---|---|---|
| | | | | | aOR | 95% CI | | aOR | 95% CI | | aOR | 95% CI | |
| 4th wave (Omicron) | Anti-N negative, no vaccination* | 607 | 16 | 7 | Ref | | | Ref | | | Ref | | |
| | Anti-N negative, partial vaccination | 95 | 1 | 0 | 0.41 | 0.05 | 3.18 | 0.73 | 0.09 | 5.89 | . | | |
| | Anti-N negative, full vaccination | 450 | 12 | 5 | 0.87 | 0.40 | 1.90 | 1.00 | 0.36 | 2.78 | 0.64 | 0.19 | 2.12 |
| | Anti-N positive, no vaccination | 4,221 | 31 | 7 | 0.29 | 0.16 | 0.54 | 0.39 | 0.18 | 0.86 | 0.15 | 0.05 | 0.46 |
| | Anti-N positive, partial vaccination | 883 | 22 | 0 | 0.97 | 0.50 | 1.92 | 1.61 | 0.72 | 3.62 | . | | |
| | Anti-N positive, full vaccination | 1,928 | 11 | 2 | 0.22 | 0.10 | 0.47 | 0.33 | 0.13 | 0.85 | 0.07 | 0.01 | 0.35 |
| | | 8,184 | 93 | 21 | | | | | | | | | |
| 3rd wave (Delta) | Anti-N negative, no vaccination | 1,672 | 114 | 42 | Ref | | | Ref | | | Ref | | |
| | Anti-N negative, partial vaccination | 152 | 12 | 2 | 1.04 | 0.55 | 1.97 | 1.43 | 0.71 | 2.89 | 0.41 | 0.10 | 1.74 |
| | Anti-N negative, full vaccination | 85 | 0 | 0 | . | | | . | | | . | | |
| | Anti-N positive, no vaccination | 4,066 | 18 | 1 | 0.08 | 0.05 | 0.13 | 0.10 | 0.06 | 0.18 | 0.02 | 0.00 | 0.11 |
| | Anti-N positive, partial vaccination | 246 | 3 | 0 | 0.18 | 0.06 | 0.58 | 0.28 | 0.09 | 0.92 | . | | |
| | Anti-N positive, full vaccination | 85 | 1 | 0 | 0.20 | 0.03 | 1.49 | 0.37 | 0.05 | 2.75 | . | | |
| | | 6,306 | 148 | 45 | | | | | | | | | |
| 2nd wave (Beta) | Anti-N negative, no vaccination | 1,385 | 38 | 16 | Ref | | | Ref | | | Ref | | |
| | Anti-N positive, no vaccination | 1,866 | 6 | 2 | 0.13 | 0.05 | 0.31 | 0.15 | 0.05 | 0.45 | 0.10 | 0.02 | 0.45 |
| | | 3,251 | 44 | 18 | | | | | | | | | |

*Logistic regression for the outcome of being a SARS-CoV-2 case, a non-severe case or having severe COVID-19 (admission/death)*

* Vaccination was assessed at time of SARS-CoV-2 diagnosis, or if no diagnosis was made, at the peak of the relevant wave. Full vaccination was defined as being >14 days after a second dose of Pfizer–BioNTech (BNT162b2).

Despite the differences in study design, our seroprevalence estimates were similar to other contemporaneous studies in South Africa and Africa [5–12]. A seroprevalence study across three provinces in South Africa, including the Western Cape, found a seroprevalence of 53% in April 2021 [7], compared to the 50.1% we found in 8 February to 21 May 2021. And in 20 September to 27 November 2021, we documented a prevalence of 67.8%, comparable to the 73.1 (95%CI 72.0–74.1) found in Gauteng Province in 22 October to 9 December 2021, with again the slightly later sampling dates in these studies most likely resulting in their marginally higher seroprevalences (5). Furthermore, we observed similar heterogeneity in the seroprevalence across demographic and social economic factors. Race remains a strong predictor of socio-economic status in South Africa, and data from blood donor samples collected in March 2022, showed that Black donors had a seroprevalence 90%, 1.5 times higher than White donors with a seroprevalence of 60% [6]. HIV infection and less affluent subdistrict of residence, proxies for lower socio-economic status, were associated with higher seroprevalence across all three waves of infection in our study. In a previous local study, these factors were also strongly associated with hospitalization and death, but at that point we could not differentiate the impact of high infection burden from other factors which might drive disease progression [23]. The current study suggests that the higher mortality seen in poorer communities was substantially driven by higher risks of infection.

The epidemic wave asymmetry across subdistricts is best illustrated by Fig 4. The high levels of seropositivity from the first wave of COVID-19 infections in Khayelitsha likely provided protection from infection and severe disease in subsequent waves, despite the second and third

waves being caused by variants of higher virulence [27]. Omicron as an immune escape variant, however, resulted in a surge of infections in the fourth wave, even in areas like Khayelitsha.

Our study suggests that official case counts in Cape Town represented only ~10% of all infections, and that the degree of under-ascertainment was highest in low-income communities. This case under-ascertainment has been observed in low- and middle-income countries and highlights the reality of unequal access to diagnosis and care [1]. In addition, this under-ascertainment hampers the international comparison of epidemiology and pandemic trajectories, which in turn impacts resource allocation for future global response to emerging pathogens. Our findings suggest that in future outbreaks, an increase in diagnostic resources should be considered for poorer communities, as surveillance based on routine diagnostic systems might not accurately reflect their disease burden. Seroprevalence data can help augment the available surveillance data and direct resources, such as personal protective equipment and treatment option, to those communities most at risk. A recent meta-analysis found that age-specific infection fatality rates of COVID-19 were in fact twice as high in developing countries, compared to high income countries, as a result of elevated transmission to older adults and inadequate health care, highlighting the need for equity in the provision of medical technologies and services [28].

This study shows that anti-N positivity, a proxy for prior exposure, was associated with protection against severe disease, even in the context of an immune escape variant like Omicron, reducing the odds of hospital admission or death by 85%. Another study from the Western Cape, using different methodology, found that prior diagnosed and documented infection protected against COVID-19 hospitalization or death with an adjusted hazards ratio of 0.28 (95% CI 0.19–0.40) over the first four COVID-19 waves, although undiagnosed prior infection was listed as a limitation for this study [27]. The protection against confirmed symptomatic infection that we found was more modest but the under-ascertainment of milder cases, particularly in the poorer communities, could have biased this estimate towards the null. This bias was further evident in the similar aORs for confirmed symptomatic infection and admission during the second wave. This was the period where testing capacity was severely constrained and there was greater bias towards testing individuals with risk factors for severe disease.

While the fourth wave saw a sharp increase in SARS-CoV-2 cases, this did not result in a concomitant increase in related mortality (see line graph in Fig 1). In line with established global literature, analyses from our setting have confirmed that the Omicron variant was associated with less severe disease when compared to the Delta variant [27,29], but the low case ascertainment rates in this study complicates this interpretation [30]. Vaccination in addition to natural infection (anti-N seropositivity) further halved the risk of severe disease in our study. A recent meta-analysis, which incorporated studies from Europe and the Americas, found that while immunity wanes with time, individuals with this type of hybrid immunity had the highest magnitude and durability of protection against severe COVID-19, with protection of 97.4% at 12 months (95%CI 91.4–99.2), compared to 74.6% (95% CI 63.1–83.5) for previous infection alone after the same time period [31]. And the combination of both vaccination and recent infection was found to confer the highest level of neutralizing antibodies against all variants, including Omicron sub-lineages [32]. This broader, more potent immune response associated with this hybrid immunity likely plays an important role in the ongoing decoupling of COVID-19 cases and severe disease in South Africa, and highlights the value of vaccination even in communities which had already achieved high levels of seropositivity.

While the linked analysis highlights how seropositivity to SARS-CoV-2 is protective for an individual in subsequent waves of COVID-19, and the ecological analyses demonstrates how it

can be advantageous for communities to have high seroprevalence rates when they face subsequent waves of COVID-19, it is important to bear in mind that seropositivity is only achieved through the risk of severe illness and death during the first infection of SARS-CoV-2 (Fig 3C).

## Limitations

This study has several limitations. We were limited to using residual samples, of mostly chronically ill patients in care in the public sector, and it is uncertain how representative they are of the general population, with regards to exposure to infection and SARS-CoV-2 testing patterns. The residual samples we used also had a higher proportion of women and older adults, compared to the general population of Cape Town. While representativity of the general population is somewhat compromised by using residual samples, this study would not otherwise be feasible to perform in our setting, and we are still able to assess the differences in seroprevalence across various population groups.

For the ecological analysis, as suburb data was incomplete, we were restricted to using the subdistrict level, even though marked heterogeneity can exist within subdistricts.

In the linked serology analysis, misclassification of both the exposure and the outcome could have occurred, as the sensitivity of the antibody assay is imperfect, and with limited SARS-CoV-2 testing, particularly at the peaks of waves, there was under-ascertainment of cases. Related to testing, those who were vaccinated were more likely to access/present for diagnostic testing, resulting in health seeking behaviour being a confounder which may partly explain the lack of protective vaccination effect for the outcome of being a case. In addition, antibody results were assessed as positive or negative only, and the quantitative antibody values, which would give an indication of waning protection with time, were not examined.

While this study did find additional protection against severe disease from vaccination in those with prior infection, it is not a formal study of vaccine effectiveness. Most of the study population were diabetic or living with HIV, and the findings cannot directly be extrapolated to the general population. With such high background seroprevalence rates in the public sector population, the small numbers of individuals who remained anti-N negative in November 2021 might have a biological reason for not seroconverting, i.e., have suboptimal immune responses. This analysis also had low numbers of seronegative, vaccinated individuals prior to the fourth wave, and since anti-N positivity was associated with strong protection against severe disease, we were underpowered to adequately assess the additional benefit of vaccination amongst those who were already seropositive.

Another unavoidable limitation of this study is survivor bias, in that only those individuals who survive their first episode of COVID-19 can have their serology tested at a later time point and found to be seropositive.

## Conclusion

There is a complex interaction over time between socio-economic status and SARS-CoV-2 infection with differing variants, as well as the protection that both prior infection and vaccination confers, that has resulted in diverse COVID-19 wave patterns and experiences of the pandemic in different communities.

Seroprevalence to SARS-CoV-2 in Cape Town increased rapidly over three epidemic waves, with poorer communities having consistently higher prevalence rates. While seropositivity was highly protective in later infections at the individual level, such protection was only achieved at the cost of high initial population COVID-19 morbidity and mortality.

In settings such as ours, where diagnostic testing capacity is limited, surveillance systems dependent on them will underestimate the true extent of an outbreak, and public health

officials need to appreciate issues around low and differential case ascertainment rates to understand the true extent and nature of the outbreak. Rapidly conducted seroprevalence studies can play an important role in augmenting this understanding.

## Author Contributions

**Conceptualization:** Hannah Hussey, Mary-Ann Davies, Emma Kalk, Andrew Boulle, Nei-yuan Hsiao.

**Data curation:** Alexa Heekes, Jamy-Lee Bam, Nesbert Zinyakatira, Mariette Smith.

**Formal analysis:** Hannah Hussey, Nesbert Zinyakatira.

**Funding acquisition:** Robert Wilkinson.

**Methodology:** Helena Vreede, Nei-yuan Hsiao.

**Supervision:** Mary-Ann Davies, Andrew Boulle, Nei-yuan Hsiao.

**Visualization:** Hannah Hussey.

**Writing – original draft:** Hannah Hussey, Nei-yuan Hsiao.

**Writing – review & editing:** Hannah Hussey, Helena Vreede, Mary-Ann Davies, Emma Kalk, Diana Hardie, Gert van Zyl, Michelle Naidoo, Erna Morden, Chad M. Centner, Jean Maritz, Jessica Opie, Zivanai Chapanduka, Hassan Mahomed, Annibale Cois, David Pienaar, Andrew D. Redd, Wolfgang Preiser, Robert Wilkinson, Andrew Boulle, Nei-yuan Hsiao.

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
