## [Decision Letter · Decision Letter 0]

20 Dec 2023

PGPH-D-23-01530

SARS-CoV-2 seroepidemiology in Cape Town, South Africa, and implications for future outbreaks in low-income communities

Dear Dr. Hussey,

Thank you for submitting your manuscript to PLOS Global Public Health. After careful consideration, we feel that it has merit but does not fully meet PLOS Global Public Health’s publication criteria as it currently stands. Therefore, we invite you to submit a revised version of the manuscript that addresses the points raised during the review process.

Please note that we have only been able to secure a single reviewer to assess your manuscript. We are issuing a decision on your manuscript at this point to prevent further delays in the evaluation of your manuscript. Please be aware that the editor who handles your revised manuscript might find it necessary to invite additional reviewers to assess this work once the revised manuscript is submitted. However, we will aim to proceed on the basis of this single review if possible.

The reviewer has provided feedback throughout your manuscript, and particularly regarding the context, description of the methods, and has provided suggestions to improve the discussion. Please ensure you address each of the reviewer's comments when revising your manuscript.

We look forward to receiving your revised manuscript.

Kind regards,

Hugh Cowley

Staff Editor

Journal Requirements:

1. Please update your online Competing Interests statement. If you have no competing interests to declare, please state: “The authors have declared that no competing interests exist.”

2. Please provide separate figure files in .tif or .eps format only and ensure that all files are under our size limit of 10MB.

Additional Editor Comments (if provided):

Reviewers' comments:

Reviewer's Responses to Questions

**Comments to the Author**

1. Does this manuscript meet PLOS Global Public Health’s publication criteria? Is the manuscript technically sound, and do the data support the conclusions? The manuscript must describe methodologically and ethically rigorous research with conclusions that are appropriately drawn based on the data presented.

Reviewer #1: Yes

2. Has the statistical analysis been performed appropriately and rigorously?

Reviewer #1: Yes

3. Have the authors made all data underlying the findings in their manuscript fully available (please refer to the Data Availability Statement at the start of the manuscript PDF file)?

Reviewer #1: Yes

4. Is the manuscript presented in an intelligible fashion and written in standard English?

Reviewer #1: Yes

5. Review Comments to the Author

Reviewer #1: Overall comments

This is a valuable paper describing seroprevalence and antibody positivity correlation with protection against severe disease in South Africa. I commend the authors for their research, which is also well situated in a global health context and providing insights on seroepidemiology for vulnerable populations.

My main major comment is to please situate context for the study for the population, sampling, and discussion where possible – i.e. describing the population demographics and landscape, vaccine availability and types available in the region over time, and any changes in testing policy over the course of the study, given the sampling date range is very broad. I would also suggest the authors consider bolstering their analyses with test and/or population adjustment for seroprevalence estimates. Please also add more comparison of your results with other literature measuring antibody positivity correlation with protection against severe disease in the discussion section.

Comments:

Abstract

Line 24- Suggest deleting ‘determinants’ or rephrasing for clarity

Line 33-34: Please include 95% confidence intervals for results if available

Could you include a short line on how you elucidated protection against severe disease and reported COVID-19 tests (what is your data source?)

Introduction:

Line 60-63: How do these excess deaths compare to non-COVID excess deaths? Important to have a comparison to make your statement on possible case under-ascertainment stronger.

Line 63: Note typo with extra space and period.

Line 65: What are the barriers to testing?

Line 70: Please elucidate elements needed to capture immune protection – i.e. quantitative antibody testing to detect antibody wane, etc. It is not just longitudinal versus cross sectional that limits these results being available. Also, for your consideration, here are a few South African serosurveys that are longitudinal, which you may wish situate the background in the context of these findings.

https://www.mdpi.com/1999-4915/14/6/1222

https://www.mdpi.com/2077-0383/12/2/529

Methods:

Line 99: Could you please give an indication of approximately how many samples were processed by NHLS over the course of the study duration?

Line 102: Please describe more information on the convenience sampling method used. I.e. did you select all samples on certain dates, first N of samples per month, etc, or were all samples included?

Lines 101-107: Were there any exclusion criteria, or a sample size calculation used? If so, please describe.

Lines 110-111: Somewhere appropriate in the paper please describe the vaccination landscape in South Africa at the times of sampling, and which vaccines were predominantly administered (N-detecting assays for prior infection will not work if the population has been largely vaccinated with a non-mRNA vax such as the Oxford-Astra Zeneca vax).

Line 112-115: Please describe the performance testing in South Africa if possible. It would also be advantageous to list the denominator for Sn and Sp testing (i.e., how many samples were used to determine the performance)?

Line 124-147: How was the vaccination status data analytically linked with the natural infection elucidation from your anti-N assay, particularly for Janssen, as recipients of a viral vector-based vaccine would produce anti-N antibodies?

Line 151-153: The sensitivity of the assay used in this paper is rather low. I would consider statistically adjusting the seroprevalence estimate by test performance.

Line 167: Please provide specific date windows for each ‘wave’ of sampling.

Results

198: Replace epidemic with pandemic

Line 211-213: This sentence is confusing, consider re-phrasing or breaking up results into different sentences

Discussion

Line 327: On this result on hybrid immunity – I would move this earlier to the top of your discussion with your other remarks on anti-N protection against severe disease, this is a higher-impact result. It would also be great to situate these results against VE or PE studies/meta-analyses – understanding this is not a VE or PE study in itself, but would be interesting to compare.

Line 342-344: If you have demographic information available, you could statistically adjust for population representativeness.

6. PLOS authors have the option to publish the peer review history of their article (what does this mean?). If published, this will include your full peer review and any attached files.

**Do you want your identity to be public for this peer review?** For information about this choice, including consent withdrawal, please see our Privacy Policy.

Reviewer #1: **Yes: **Mairead Whelan

---

## [Decision Letter · Decision Letter 1]

11 Jul 2024

SARS-CoV-2 seroepidemiology in Cape Town, South Africa, and implications for future outbreaks in low-income communities

PGPH-D-23-01530R1

Dear Dr Hussey,

We are pleased to inform you that your manuscript 'SARS-CoV-2 seroepidemiology in Cape Town, South Africa, and implications for future outbreaks in low-income communities' has been provisionally accepted for publication in PLOS Global Public Health.

Best regards,

Leeberk Raja Inbaraj, MD

Academic Editor
